# The Association between Nursing Students’ Happiness, Emotional Intelligence, and Perceived Caring Behavior in Riyadh City, Saudi Arabia

**DOI:** 10.3390/healthcare12010067

**Published:** 2023-12-28

**Authors:** Nadiah A. Baghdadi, Chandrakala Sankarapandian, Judie Arulappan, Murad H. Taani, Julia Snethen, Shaherah Yousef Andargeery

**Affiliations:** 1Nursing Management and Education Department, College of Nursing, Princess Nourah bint Abdulrahman University, P.O. Box 84428, Riyadh 11671, Saudi Arabia; nabaghdadi@pnu.edu.sa; 2Medical Surgical Nursing Department, College of Nursing, Princess Nourah bint Abdulrahman University, P.O. Box 84428, Riyadh 11671, Saudi Arabia; csshenbagathammal@pnu.edu.sa; 3Department of Maternal and Child Health, College of Nursing, Sultan Qaboos University, Al Khoudh, Muscat 123, Oman; judie@squ.edu.om; 4School of Nursing, University of Wisconsin, Milwaukee 1921 East Hartford Avenue, Milwaukee, WI 53211, USA; mhtaani@uwm.edu (M.H.T.); julia@uwm.edu (J.S.)

**Keywords:** caring behaviors, caring behavior inventory, caring experience, emotional intelligence, happiness, nursing care, nursing education, nursing practice, nursing students

## Abstract

Background: Caring behavior is a major focus of the nursing profession and an important dimension of nursing practice that sets nurses apart from other healthcare professionals. Effective patient-centered care requires ensuring nurses have the emotional intelligence and happiness to address the daily demands of practice. The purpose of this study is to examine the emotional intelligence and happiness among nursing students and their relationship with caring behaviors. Methods: A cross-sectional, descriptive correlational study was conducted on nursing students (*n* = 363) from Riyadh, Kingdom of Saudi Arabia, via an online survey. Measures include demographic data survey, Oxford Happiness Questionnaire, Trait Emotional Intelligence Questionnaire, and Caring Behaviors Inventory scale. Descriptive and multiple regression analyses were conducted for this study. Results: Nursing students reported their highest degree of caring was in terms of ‘respectful differences to others’, while their lowest was in ‘knowledge and skills’. Emotional intelligence and happiness were significant predictors of caring behaviors and explained the variance in assurance of human presence (17.5%), knowledge and skills (17.5%), respectful differences to others (18%), and positive connectedness (12.9%). In the final regression model, emotional intelligence and happiness were significant predictors of caring behaviors and explained 19.5% of the variance. Conclusions: Emotional intelligence and happiness among nursing students were found to be important factors to improve their caregiving behaviors. Therefore, nursing educators should consider integrating emotional intelligence and happiness interventions for students into their curriculum.

## 1. Background

Caring behavior is integral to the nursing profession and an important dimension of nursing practice that makes nurses unique. Ref. [1] described caring as ‘a growing art in nursing’, and ‘caring science informs and serves as the moral, philosophical, theoretical, and foundational starting point for nursing education, patient care, research, and even administrative practices’ ([1] p. 16). Caring is an essential human need that nurses carry out to provide physical and emotional comfort and improve the wellbeing of the patients [2,3]. Nurses provide caring behavior when interacting with patients and families, which is influenced by their education, individual characteristics, qualities, cultural backgrounds, and the environment [4]. Caring behavior is a complex social interaction that improves by commitments in mutual relationships [5]. Building trust was found to be integral for providing care and allowing nurses to accept the care. According to [6], caring behavior among nurses enhances the quality of nursing care and patient safety. It is an essential quality for nurses in clinical settings to provide holistic and patient-centered care, as such it is a critical quality for nursing students to develop as they transition into the healthcare profession. Thus, effectively developing and promoting caring attributes and characteristics within nursing curricula and among nursing students is a key element of nursing education.

Nurses and nursing students are educated and trained to provide the highest level of care to improve their patients’ health outcomes and enhance the overall quality of healthcare services. However, several factors are associated with caring behaviors of frontline nurses and nursing students, such as the demands of nursing education and profession [7]. Research has also shown relationships between happiness and depression, self-esteem, and interpersonal relationships among nursing students, which all may be associated with caring behaviors. Happiness is a subjective experience and a stable emotional state characterized by pleasant feelings and gratification, a sense of overall wellbeing, joy, prosperity, and personal growth [8,9]. Happiness is linked to reduced stress levels, increased resilience, greater self-confidence, and positive emotions and attitudes [10,11]. It has been reported that happiness is associated with attitudes and behaviors [12]. Nurses who experience a sense of happiness are better equipped to provide compassionate and patient-centered care. When nurses are happy and fulfilled in their roles, they are more likely to exhibit empathy, be active listeners, and create a healing environment where patients feel valued and supported. In contrast, nursing is considered a high-pressure, stressful, and challenging profession, and nursing students, similar to nurses, are expected to meet the demanding requirements of nursing education, have limited opportunities to express their opinions and make decisions, work long shifts, and must adhere to rules and regulations [13]. In clinical settings, nursing students regularly encounter individuals who are experiencing suffering, grief, and distress. The stress experienced by nursing students not only impacts them, but also affects other nurses and the overall efficiency and quality of care within the healthcare system [13,14]. Evidence shows that nursing students have high levels of stress and low levels of happiness [9], which may impact their caring behaviors.

Emotional intelligence refers to an individuals’ ability to understand, evaluate, manage, and effectively utilize their emotions when thinking or taking actions [15]. Nursing students with effective emotional intelligence demonstrate greater adaptability and optimistic attitudes, which results in better relationships and improved values [16]. Thus, emotional intelligence is important in clinical settings. Healthcare professionals with high levels of emotional intelligence tend to be more compassionate, empathic, resilient, and able to manage emotions in others, making them more capable and effective in caring for themselves and their patients and families [17]. Hence, emotional intelligence could be necessary to achieve the daily demands of nursing and to provide sufficient patient-centered care. Emotional intelligence can also play a vital role in improving performance and building effective relationships. Research has shown that both patient-related outcomes and nursing students’ outcomes are linked to emotional intelligence and that the quality of care is influenced by the emotional intelligence of nurses [14].

Although research has been conducted on emotional intelligence and happiness, no studies have been conducted in the Kingdom of Saudi Arabia (KSA) that describe emotional intelligence and happiness and examine their relationships to caring behaviors among Saudi nursing students. We found one study on caring behavior among Saudi nursing students, but it did not examine emotional intelligence and happiness [18]. Considering the importance of emotional intelligence and happiness and due to the lack of studies on both emotional intelligence and happiness and their relationship to caring behavior among Saudi nursing students, this study aimed to describe both emotional intelligence and happiness and examine their relationship with the perceived caring behaviors among Saudi nursing students. We hypothesized that emotional intelligence and happiness to be associated with perceived caring behaviors among Saudi nursing students.

## 2. Methods

### 2.1. Design, Setting, and Sample

A cross-sectional, descriptive correlational design was used to examine the relationship between nursing students’ happiness and emotional intelligence and their perceived caring behaviors. A purposive sampling was used in this study and the questionnaires were distributed to nursing students in a nursing college located in Riyadh, the capital of KSA. The inclusion criteria included: (1) nursing students who were studying for a bachelor’s degree, (2) were able to understand written English, (3) completed at least one clinical course, and (4) voluntarily consented to participate in the study. To ensure an adequate sample size for statistical power, a power analysis was calculated using G*Power. For the 6 predictor variables and the multiple linear regressions on the nurses’ caring behaviors, a sample of 146 student nurses was needed with a 0.15 effect size, 0.05 margin of error, and 95% statistical power.

### 2.2. Measurements

Demographic data collection included age, gender, grade point average (GPA), and year of study. Measures to examine the independent and dependent variables of the study included questionnaires on emotional intelligence, happiness, and caring behaviors.

Emotional Intelligence. The Trait Emotional Intelligence Questionnaire Short Form (TEIQue-SF) [19] was used to measure the global trait of emotional intelligence. The 30 items included a Likert-style response option ranging from 1 (completely disagree) to 7 (completely agree). The scale had four factors: wellbeing, self-control, emotionality, and sociability. The model of questions was as follows: (1) I usually find it difficult to regulate my emotions; (2) I can deal effectively with people; (3) On the whole, I’m able to deal with stress, etc. Completely agree responses indicated a higher sense of wellbeing, meaning the person was satisfied with life. Higher self-control scores indicate the participant had a greater ability to manage and regulate external pressures effectively. A high emotionality score indicates that the person had emotion-related skills including recognizing internal emotions, perceiving emotions in others, and expressing their emotional states. Higher sociability scores suggested effective listening communication skills, with a focus on an individual’s social relationships and influence and evaluating a person’s impact in various social contexts [20]. A global trait emotional intelligence score was calculated by adding up the item scores and then dividing the sum by the total number of items. An internal consistency score of 0.81 was reported for this questionnaire with a test−retest reliability of 0.86 for the total score [21]. The Cronbach’s alpha coefficient for the TEIQue-SF questionnaire was 0.85 in this study.

Happiness. The Oxford Happiness Questionnaire (OHQ) was administered to measure participants’ happiness. The questionnaire consisted of 29 items that measure inner happiness using a 6-point response scale: 1 = strongly disagree, 2 = moderately disagree, 3 = slightly disagree, 4 = slightly agree, 5 = moderately agree, and 6 = strongly agree. The pattern of questions was: (1) I feel that life is very rewarding; (2) I am well satisfied about everything in my life; (3) I do not find it easy to make decisions, etc. The negative questions scores were provided a reversed score.

The OHQ scale has an internal reliability score of 0.90 and a test−retest reliability score of 0.78. For this study, the Cronbach’s alpha coefficient of the OHQ scale was 0.76.

Caring behaviors. The Caring Behaviors Inventory (CBI-24) was used to assess the caring behaviors among nursing students [22]. CBI is 24 items and is divided into four subscales, namely assurance, knowledge and skill, respect, and connectedness, with Likert-scale responses of 1 = never, 2 = almost never, 3 = occasionally, 4 = usually, 5 = almost always, 6 = always. The examples of questions for caring behaviors was as follows: (1) showing concern for the patient; (2) attentively listening to the patient; and (3) giving health teaching to the patient etc. The mean scores were calculated, with a higher mean score indicating higher caring behaviors reported by nursing students. CBI-24 demonstrated a test–retest reliability score of 0.82 with a Cronbach’s alpha value of 0.95 [22]. The Cronbach’s alpha coefficient for the CBI-24 was 0.96 in this study.

### 2.3. Ethical Considerations

Institutional Review Board (IRB) approval (Registration Number: 20-0494) was obtained from Princess Nourah bint Abdulrahman University IRB prior to the initiation of the investigation. Student nurses were provided with detailed information about the study and participation prior to obtaining consent. All of the students were informed that participation in the study was entirely voluntary and the survey was anonymous. They were also assured that the study also had no potential for harm regarding their participation. They were also assured that the data would be secured and only be used for research purposes. They were reassured that they could abstain from answering any questions that made them feel uncomfortable and withdraw from the study at any time without any penalty.

### 2.4. Data Collection

Data were collected by sending a link to potential participants for accessing the online survey. Students who accessed the survey link and agreed to participate received written instructions provided online on the process for responding to the survey questions. The online survey link was opened from November 2021 and July 2022.

### 2.5. Data Analysis

SPSS^®^ Version 28 was used to conduct the descriptive and inferential statistical analysis. Demographic data and the variables of emotional intelligence, happiness, and caring behaviors were initially analyzed by conducting descriptive statistics. Multiple regression models were created on the four subscales of caring behaviors (knowledge and skills, assurance of human presence, respectful difference of others, and positive connectedness), and one multiple regression model was created using the sum score from the caring behaviors scale. Emotional intelligence and happiness were used as predictors and age, gender, study year, and GPA were used as the control variables. All of the statistical assumptions for each regression model were examined, including multicollinearity and residual normality, and all assumptions were met. Gender and study year predictors were dummy-coded before including them in the regression models. A *p*-value of <0.05 was used for statistical significance. A total of 376 participants completed the survey. Participants (*n* = 13) with missing data were excluded from the analysis and the data of 363 participants were used in the analyses.

## 3. Results

### 3.1. Results of the Descriptive Analyses of the Study Variables

The sample characteristics are depicted in Table 1. Participants’ mean age was 20.11 years old (SD = 1.33) with the majority being female (*n* = 346, 95.3%), and a mean GPA of 3.5 (SD = 0.81). Around half of the participants (*n* = 184, 50.7%) were in the third year of their nursing program. In terms of overall emotional intelligence level, participants demonstrated a high level of emotional intelligence with a mean score of 4.34 (SD = 0.57). Furthermore, the highest mean of 4.83 (SD = 0.91) was recorded in the subscale ‘wellbeing’, followed by self-control (M = 4.28, SD = 0.76) and sociability (M = 4.18, SD = 0.77). The subscale ‘emotionality’ received the lowest mean of 4.14 (SD = 0.75). Moreover, the mean score of happiness was 4.06 (SD = 0.75), indicating a moderate level of happiness. In terms of the students’ caring behaviors, the average score for caring behaviors was 5.04 (SD = 0.89). The highest mean of 5.09 (SD = 1.03) was recorded in the subscale ‘respectful differences to others’, followed by ‘assurance of human presence’ with a mean of 5.04 (SD = 0.90), ‘positive connectedness’ with a mean of 5.02 (SD = 0.99), and ‘knowledge and skills’ with a mean of 5.02 (SD = 1.02).

Among the demographic characteristics, there were statistically significant differences in the total mean score of emotional intelligence (*p* = 0.003), happiness (*p* = 0.014), and caring behaviors (*p* = 0.013) based on GPA. Those with GPA ≥ 3.5 demonstrated higher emotional intelligence, happiness, and caring behaviors scores. There were also statistically significant differences in happiness based on study year (*p* = 0.001). Students in their fourth and fifth years of study demonstrated significantly higher happiness scores compared with those in their first year of study. The differences in emotional intelligence, happiness, and caring behaviors based on GPA, sex, and study year are depicted in Table 2.

### 3.2. Correlation between the Study Variables

In the correlation analysis (Table 3), moderate positive statistically significant correlations were found between emotional intelligence and caring behaviors (r = 0.385, *p* < 0.001), as well as between happiness and caring behaviors (r = 0.337, *p* < 0.001). A strong positive statistically significant correlation was found between emotional intelligence and happiness (r = 0.528, *p* < 0.001).

### 3.3. Predictors of Caring Behaviors

Emotional intelligence and happiness were significant predictors of all subscales of caring behaviors. Specifically, emotional intelligence and happiness explained 17.5% of the variance in assurance of human presence and 17.5% of the variance in professional knowledge and skills (Table 4).

Emotional intelligence and happiness also explained 18% of the variance in respectful differences to others and 12.9% of the variance in positive connectedness (Table 5).

In the final regression analysis (Table 6), the model explained 19.5% of the variance in the mean caring behaviors score (F = 14.372, *p* < 0.001). Both emotional intelligence (b = 0.411, *p* < 0.001) and happiness (b = 0.349, *p* < 0.001) were statistically significant predictors of the mean caring behaviors score.

## 4. Discussion

Caring is the essence and core principle of nursing. Care and compassion are prerequisites for nursing practice [7,18]. Nurses play a central role in delivering holistic patient care by addressing the patient’s physical, psychological, social, and spiritual needs, enabling them to deal with their illness, restore their balance, and improve their lives [8,23]. Thus, the nursing practice is centered around caring and the caring behavior of nurses, which has a definite impact on the quality of patient care and patient satisfaction [7,18]. Therefore, nurses must demonstrate caring behaviors to create a therapeutic relationship with patients. Nurses are expected to have emotional intelligence characterized by self-awareness, social awareness, self-management, and relationship management that improves the quality of nursing care [3]. In addition, nurses’ happiness becomes very important in nursing practice, as unhappiness may affect nurses’ ability to care for the patients. Thus, the happiness and emotional intelligence of the nurses may have an impact on the caring behavior of nurses. Very little is known about the relationship between these factors. Therefore, our study was conducted among Saudi nursing students and found emotional intelligence and happiness to be predictors of caring behaviors and its four components (i.e., assurance of human presence, professional knowledge and skills, respectful differences to others, and positive connectedness). The findings did not change after controlling for any possible effects of age, sex, study year, and GPA.

No studies found in the literature investigated the relationship between Saudi nursing students’ caring behaviors and their emotional intelligence or happiness. Nevertheless, studies conducted in other countries have similarly reported a relationship between nurses’ emotions and the quality of care they provide to their patients. Emotional intelligence positively affects the quality of nursing care [14]. Likewise, emotionally intelligent nurses’ experience increased job satisfaction, reduced stress levels, and burnout. Specifically, investigators found that the emotional intelligence skills of nursing students improved their decision making, impacted clinical learning situations, and facilitated student nurse transition into practice [14,24,25,26], which were related to the ‘professional knowledge and skills’ domain of emotional intelligence. Our findings among Saudi nurses aligned with these results by showing a high mean score on the ‘level of knowledge and skills’ domain, which was also predicted by emotional intelligence. This shows that emotional intelligence is central to nursing practice and it has the potential to influence the decision making, critical thinking, quality of patient care, and overall wellbeing of practicing nurses [27].

Our study found emotional intelligence to be associated with the ‘respectful differences to others’ domain of caring behaviors, which recorded the highest domain mean score by our participants. Prior investigations similarly reported that emotional intelligence skills and competencies play a critical role in enabling nurses to develop empathic and therapeutic relationships with patients and provide explicit notions of support and caring within human relationships [14,25,26]. The level of emotional intelligence in nurses has been found to be positively correlated with increasing their abilities to demonstrate caring behaviors [28]. In the study conducted by [29], it was found that when the overall emotional intelligence increased, the caring behaviors provided by nurses increased as well. These results indicate that emotional intelligence is a vital factor in nurses’ caring behaviors. Nursing students who possess high emotional intelligence are able to communicate with their patients on an emotional level, understand their personal needs, and demonstrate caring behavior.

Furthermore, emotional intelligence was found, in this study, to be associated with the ‘assurance of human presence’ and ‘positive connectedness’ domains of caring behaviors in Saudi nursing students. High emotionally intelligent nurses who have high self-control against criticism and emotional awareness and sensitivity are more adept at showing concern for their patients in a respectful way and able to communicate their concerns with them [27]. Nurses who are emotionally intelligent are also able to trust relationships and mutual understanding and develop positive connectedness [27], all of which is reflected in the ‘assurance of human presence’ and ‘positive connectedness’ domains of caring behaviors among Saudi nursing students. Furthermore, our participants reported a moderate level of emotional intelligence, indicating their understanding of their ability to recognize, evaluate, and manage emotions in themselves and their patients. Yet, this moderate level of emotional intelligence suggests that new strategies to address wellbeing, self-control, sociability, and emotionality should be evaluated to improve the overall emotional intelligence among Saudi nursing students. These findings call for policy development to develop strategies and interventions to maintain a high level of emotional intelligence among nurses [16,29]. Moreover, nurse managers and hospital administrators are encouraged to organize frequent training sessions on enhancing the emotional intelligence of nurses.

The key to an effective nursing profession lies in happiness. Nurses who experience happiness in their profession enjoy their professional roles and duties and are more likely to be creative in providing holistic care to their patients. In the present study, Saudi nursing students reported a moderate level of happiness, which was consistent with previous research findings on happiness among nursing and medical sciences students [8,23]. Our study also revealed that happiness is positively associated with the student nurses caring behaviors. A previous study also showed that student nurses’ happiness significantly affects their effectiveness and caring behavior [30]. Happiness has been linked to various aspects of one’s life, including improved physical and mental health, emotional balance, self-concept, social relationships, self-attitudes, enthusiasm for helping others, decision making, and creativity [31]. Additionally, happiness positively impacts the nurses’ overall organizational performance, including productivity, nursing services, and patient satisfaction [32]. Furthermore, individual and organizational predictors determine the happiness of nurses [8,9]. However, investigators found that nursing students often face stressful periods during their studies, in addition to factors such as life satisfaction, psychological wellbeing, and quality of life, which influence happiness among nursing students [9,13,30,33].

Thus, interventions are recommended to increase the happiness of nurses, their mental health and quality of life, and the quality of care of their patients [8,9]. Our study has shown that emotional intelligence and happiness are predictors of caring behavior in nursing students. Therefore, offering interventions that will improve the happiness and emotional intelligence of nursing students are of the utmost importance. Ultimately, these interventions will improve the wellbeing of nurses and the quality of care delivered to the patients.

Our study has several limitations that warrant careful consideration when interpreting the findings. A cross-sectional correlational design was used for this study, so cause-and-effect relationships cannot be determined. As all participants were Saudi nursing students and recruited from one institution in one city in Saudi Arabia, the findings from this study have limited generalizability to other institutions and cities in Saudi Arabia. Furthermore, the sample predominantly comprised female nurses and, as a result, caution should be exercised when attempting to generalize the findings beyond the specific demographic characteristics of the study sample. Finally, using self-reported scales may introduce some bias as there is always the chance that participants responses were influenced by societal expectations.

### Implications of the Study

To the best of our knowledge, this study is the first to examine the relationship between emotional intelligence and happiness, and caring behaviors among Saudi nursing students. The findings from this study demonstrate that emotional intelligence and happiness played a role in the caring behaviors of Saudi nursing students. Nursing educators should be aware of the need for emotional intelligence and happiness and the impact they can have on caring behaviors in order to adapt their teaching methodologies accordingly. Despite the importance of integrating emotional intelligence and happiness and related concepts such as life satisfaction into the nursing curriculum, limited information was available in the literature on the incorporation of emotional intelligence and happiness into nursing curricula and programs [34,35,36]. Thus, implementing ongoing educational training and investigating strategies to develop high levels of emotional intelligence, happiness, and caring behaviors among nursing students are warranted. For instance, elective courses within the nursing curricula could be developed to improve emotional intelligence and happiness, and ultimately enhance caring behavior. Furthermore, it would be beneficial to initiate faculty development programs to equip nursing educators with the tools and skills necessary to integrate both emotional intelligence and happiness in nursing education. Finally, larger studies with a national sample of Saudi nursing students, including a larger proportion of male students, are needed to confirm the study findings, draw more comprehensive conclusions, and compare findings based on sex differences, to enhance the generalizability of the findings.

## 5. Conclusions

Fostering a sense of caring among Saudi nursing students is crucial for meeting the nursing moral and ethical duties of nursing practice and enhancing caring behaviors. Developing emotional intelligence and happiness among Saudi nursing students can serve as a strategy to enhance their caregiving behaviors. Nursing educators should also recognize the impact of emotional intelligence and happiness on caregiving behaviors among nursing students. Thus, nursing educators should implement strategies to promote emotional intelligence and happiness, integrating them as transformational learning methods in nursing education to improve caring behaviors among Saudi nursing students.

## Figures and Tables

**Table 1 healthcare-12-00067-t001:** Sample characteristics.

Characteristic	Mean (SD)	Range	*n* (%)
Gender			
Men			17 (4.7)
Women			346 (95.3)
Study Year			
1st year			16 (4.4)
2nd year			104 (28.7)
3rd year			184 (50.7)
4th year			59 (16.2)
Age	20.11 (1.33)	18–24	
GPA	3.50 (0.81)	2.4–5	
≥3.5			181 (49.9)
<3.5			182 (50.1)
Overall Emotional Intelligence	4.34 (0.57)	3.03–6.60	
Wellbeing	4.83 (0.91)	2.33–7.00	
Self-Control	4.28 (0.76)	2.33–7.00	
Emotionality	4.14 (0.75)	2.00–6.63	
Sociability	4.18 (0.77)	1.83–7.00	
Overall Oxford Happiness Inventory	4.06 (0.75)	2.55–5.69	
Caring Behaviors Inventory	5.04 (0.89)	2.00–6.00	
Assurance of Human Presence	5.04 (0.90)	1.75–6.00	
Professional Knowledge and Skills	5.02 (1.02)	2.00–6.00	
Respectful Differences to Others	5.09 (1.03)	2.00–6.00	
Positive Connectedness	5.02 (0.99)	1.40–6.00	

**Table 2 healthcare-12-00067-t002:** Mean differences in study variables based on demographics.

Variables	Emotional Intelligence		Happiness		Caring Behaviors	
	Test	*p*	Test	*p*	Test	*p*
GPA	t = −2.955 *	0.003	t = −2.459 *	0.014	t = −2.500 *	0.013
Gender	t = 1.425	0.172	t = −0.250	0.805	t = −0.054	0.957
Study year	F = 1.063	0.375	F = 4.521 *	0.001	F = 1.557	0.185

* Statistically significant at *p* ≤ 0.05.

**Table 3 healthcare-12-00067-t003:** Correlation matrix between the study variables (*n* = 363).

	1	2	3
**1. Emotional Intelligence**	-		
**2. Happiness**	0.528 *	-	
**3. Caring Behavior**	0.385 *	0.337 *	-

*: Statistically significant at *p* ≤ 0.001.

**Table 4 healthcare-12-00067-t004:** Results of the multiple regression analyses on the nurses’ caring behaviors subscales assurance of human presence and professional knowledge and skills (*n* = 363).

	Assurance of Human Presence				Professional Knowledge and Skills			
Predictors	b	β	*p*	95% CI	b	β	*p*	95% CI
Age	−0.026	−0.039	0.541	−0.110–0.058	0.019	0.024	0.695	−0.075–0.112
Gender	0.134	0.032	0.525	−0.281–0.550	−0.065	−0.014	0.781	−0.528–0.397
Study year	0.074	0.073	0.250	−0.052–0.200	0.032	0.028	0.650	−0.108–0.173
GPA	0.048	0.044	0.380	−0.060–0.156	0.021	0.017	0.735	−0.100–0.141
Emotional intelligence	0.014	0.267	<0.001 *	0.008–0.020	0.014	0.240	<0.001 *	0.008–0.021
Happiness	0.090	0.164	0.006 *	0.003–0.015	0.015	0.242	<0.001 *	0.008–0.022
R^2^ (Adjusted R^2^) = 0.175, F (6, 356) = 11.083 *, *p* < 0.001 *					R^2^ (Adjusted R^2^) = 0.175, F (6, 356) = 13.787 *, *p* < 0.001 *			

* Statistically significant at *p* ≤ 0.05.

**Table 5 healthcare-12-00067-t005:** Results of the multiple regression analyses on the nurses’ caring behaviors subscales respectful differences to others and positive connectedness (*n* = 363).

	Respectful Differences to Others				Positive Connectedness			
Predictors	b	β	*p*	95% CI	b	β	*p*	95% CI
Age	0.021	0.027	0.660	−0.073–0.116	−0.041	−0.055	0.388	−0.134–0.052
Gender	0.138	0.028	0.561	−0.329–0.606	0.133	0.029	0.569	−0.327–0.593
Study year	0.062	0.054	0.391	−0.080–0.204	0.109	0.098	0.124	−0.030–0.249
GPA	0.065	0.052	0.295	−0.057–0.187	0.002	0.002	0.971	−0.118–0.122
Emotional intelligence	0.013	0.216	<0.001 *	0.006–0.020	0.014	0.237	<0.001 *	0.007–0.020
Happiness	0.015	0.237	<0.001 *	0.008–0.022	0.011	0.182	0.002 *	0.004–0.018
R^2^ (Adjusted R^2^) = 0.180, F (6, 356) = 12.985 *, *p* < 0.001 *					R^2^ (Adjusted R^2^) = 0.129, F (6, 356) = 9.955 *, *p* < 0.001 *			

* Statistically significant at *p* ≤ 0.05.

**Table 6 healthcare-12-00067-t006:** Results of the multiple regression analyses on the nurses’ caring behaviors total score (*n* = 363).

	b	β	*p*	95% CI
Age	−0.007	−0.011	0.857	−0.089–0.074
Gender	0.095	0.023	0.642	−0.308–0.499
Study year	0.069	0.068	0.268	−0.053–0.192
GPA	0.032	0.029	0.553	−0.074–0.138
Emotional intelligence	0.411	0.264	<0.001 *	0.236–0.587
Happiness	0.349	0.223	<0.001 *	0.172–0.526
R^2^ (Adjusted R^2^) = 0.195, F (6, 356) = 14.372 *, *p* < 0.001 *				

* Statistically significant at *p* ≤ 0.05.

## Data Availability

The datasets generated and/or analyzed during the current study are not publicly available due to data privacy, but are available from the corresponding author upon reasonable request.

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
