# Peer review of "The Association between Nursing Students’ Happiness, Emotional Intelligence, and Perceived Caring Behavior in Riyadh City, Saudi Arabia"

_healthcare, 2023, doi:10.3390/healthcare12010067_

Round 1

Reviewer 1 Report

Comments and Suggestions for Authors

Shaherah Yousef Andargeery et al. submitted to Healthcare an article focusing to the nursing students’ happiness, the emotional intelligence and the impact in the perceived caring behavior.

This manuscript is limited in its organization and lacks numerous aspects of detail, which need to be better addressed before any reevaluation.

The details are below:

- Please include in the title the territorial area in which the study was conducted;

- L 150: clarify in detail what "a sample of approximately 146 student nurses" means. What is the exact reference population (denominator)?

- L200: “Institutional Review Board approval (Registration Number: H-01- 201 R-059) was obtained from xxxxx...” What does XXX means? The same for line 208 : “The online survey 209 link was opened from xxxx 26, 202xxx, to xxxx 5, 202xxx” ?

- Specify in detail the time frame in which this study has been conducted.

- The discussion needs to be rewritten and better implemented, discussing your results with critical thinking. Furthermore, the aspects to be improved for the future must emerge (e.g. better training study plan, targeted training, community of practice workshops), useful for developing emotional intelligence. Furthermore, compare your results in depth with the studies present in the scientific literature and regarding other types of healthcare workers.

- Add the Author's Contribution section, to detail the role of all the Authors in this manuscript.

- Indicate all references in the text, according to the Instructions for the Authors of this Journal. Likewise regarding the references at the end of the manuscript, which must be listed in progressive order.

- There are numerous errors in the English language, which must be fully corrected with meticulousness and attention.

Comments on the Quality of English Language

Extensive editing of English language required

Author Response

Please include in the title the territorial area in which the study was conducted

Thanks for your comment. We added: “Riyadh, the capital of KSA”

L 150: clarify in detail what "a sample of approximately 146 student nurses" means. What is the exact reference population (denominator)?

The Power analysis revealed the need for 146 participants. We revised the sentence to improve clarity. It reads: A sample of 146 student nurses was needed with a 0.15 effect size, 0.05 margin of error, and 95% statistical power.

L200: “Institutional Review Board approval (Registration Number: H-01- 201 R-059) was obtained from xxxxx...” What does XXX means? The same for line 208 : “The online survey 209 link was opened from xxxx 26, 202xxx, to xxxx 5, 202xxx” ?

Thanks for your comment. We blinded the name of IRB. We added: Princess Nourah bint Abdul Rahman University. We also added the duration of the study: from November 2021 and July 2022.

Specify in detail the time frame in which this study has been conducted.

We added this information: from November 2021 and July 2022.

The discussion needs to be rewritten and better implemented, discussing your results with critical thinking. Furthermore, the aspects to be improved for the future must emerge (e.g. better training study plan, targeted training, community of practice workshops), useful for developing emotional intelligence. Furthermore, compare your results in depth with the studies present in the scientific literature and regarding other types of healthcare workers.

Thank you for your valuable input. The discussion section was revised.

The points that you mentioned that is related to the training can be found under the implications of the study.

Add the Author's Contribution section, to detail the role of all the Authors in this manuscript.

We added an Author Contribution section at the end of the manuscript

Indicate all references in the text, according to the Instructions for the Authors of this Journal. Likewise regarding the references at the end of the manuscript, which must be listed in progressive order.

All references were revised and in-text citations were checked and listed based on the APA formatting

There are numerous errors in the English language, which must be fully corrected with meticulousness and attention.

Thanks for your comments. The whole manuscript was edited by an English native speaker nursing faculty

Reviewer 2 Report

Comments and Suggestions for Authors

1. The study is cross-sectional, therefor, such words as impact, influence, etc. should be eliminated from the paper. Please check the whole paper.

2. Please add more keywords (up to 10) and sort them alphabetically.

3. Please provide and justify your hypotheses.

4. Please provide examples of statements of questionnaires used.

5. Lines 201: xxxxx - what does it mean?

6. Lines 209: 209 link was opened from xxxx 26, 202xxx, to xxxx 5, 202xxx. - what does it mean?

7. Long paragraphs are undesirable.

8. Table: indicate range for age and GPA.

9. Lines 247-254: Indicate clearly between what groups were statistically significant differences. What groups: higher, lower? It is unclear. Table 2 does not provide these data.

10. How was gender (and other nominal variables) coded in MRA? Please indicate in the notes. Durbin-Watson statistic? Tolerance, VIF? Indicate df in regression models.

11. β = Beta. b is not the same as β. Please clarify where you have β (std. beta coeff.), and where just b (unst. coeff).

12. Provide a correlation matrix between all study variables.

13. Internal consistency reliability coefficients as well as M (SD) should be be provided in a table with a correlation matrix.

14. Lines 406-407: Indicate clearly a name of ethical committee.

15. The authors assessed the level of EI (i.e., moderate), but based on what? Do you have norms for these questionnaires? Please compare your data with other cultural samples.

The paper should be clarified as the results in many cases are unclear for readers.

Author Response

The study is cross-sectional, therefor, such words as impact, influence, etc. should be eliminated from the paper. Please check the whole paper.

Changes are made based in the manuscript 

Please add more keywords (up to 10) and sort them alphabetically.

Thanks for your comment. We added the following: Caring Attributes, Caring behavior inventory, Caring experience. Emotional Intelligence; Happiness; Nursing care; Nursing education, Nursing Practice, Nursing students.

Please provide and justify your hypotheses

Thanks for your comment. We added our hypothesis: We hypothesized that emotional intelligence and happiness to be associated with perceived caring behaviors among Saudi nursing students.

Please provide examples of statements of questionnaires used

Sample of 3 questions were added under the tools. The questionnaires are attached as a supplemental document.

Lines 201: xxxxx - what does it mean?

Thanks for your comment. We blinded the name of IRB. We added Princess Nourah Bint Abdulrahman University

Lines 209: 209 link was opened from xxxx 26, 202xxx, to xxxx 5, 202xxx. - what does it mean?

Thanks for your comment. We also added the duration of the study: from November 2021 and July 2022

Long paragraphs are undesirable.

Changes are made

Table: indicate range for age and GPA.

We added the range for both age and GPA: 8-24 age and 2.4-5 GPA

Lines 247-254: Indicate clearly between what groups were statistically significant differences. What groups: higher, lower? It is unclear. Table 2 does not provide these data.

Thanks for your valuable comment. We added more clarification as the following: Among demographic characteristics, there were statistically significant differences in the total mean score of emotional intelligence (p=.003), happiness (p=.014), and caring behaviors (p=.013) based on GPA. Those with GPA ≥3.5 demonstrated higher emotional intelligence, happiness, and caring behaviors scores. There were also statistically significant differences in happiness based on study year (p=.001). Students in their fourth and fifth years of study demonstrated significantly higher happiness scores compared to those in their first year of study.

How was gender (and other nominal variables) coded in MRA? Please indicate in the notes. Durbin-Watson statistic? Tolerance, VIF? Indicate df in regression models.

Thanks for your comment. We have included the following information: All statistical assumptions for each regression model were examined including multicollinearity and residual normality, and all assumptions were met. Gender and study year predictors were dummy-coded before including them in the regression models. We also added the df in regression models: “F (6, 356)”

β = Beta. b is not the same as β. Please clarify where you have β (std. beta coeff.), and where just b (unst. coeff).

Thank you for your comment. We revised the tables and text to reflect the correct information.

Provide a correlation matrix between all study variables.

The correlation matrix was added with its interpretation.

Lines 406-407: Indicate clearly a name of ethical committee.

Thanks for your comment. We added: The Princess Nourah bint Abdulrahman University IRB approved the study. IRB approval No. 20-0494.

The authors assessed the level of EI (i.e., moderate), but based on what? Do you have norms for these questionnaires? Please compare your data with other cultural samples

While there are no established norms for these questionnaires, we adopted language consistent with existing literature. Our approach involved comparing our results with those reported in the literature (Amani Nezhad et al., 2022 and Ebadi et al., 2017) and describing them in a manner consistent with the existing body of work.

The paper should be clarified as the results in many cases are unclear for readers.

Changes are made

Reviewer 3 Report

Comments and Suggestions for Authors

Very interesting topic at the moment, well justified and well supported by scientific evidence in the introduction. The topic due to the positive impact it has on the academic life of nursing students and on the practice of quality nursing care.

Overall, the work is well written, with adequate scientific and research language and well-founded and despite the scarcity of research in the area, with good support. References prior to 2013 (which appear in the introduction) do not benefit the work and can be removed with the exception of those associated with the scales used (2002 and 2006). There are some small details in the text that have to be revised, due to lack of text or information (marked lines for the authors, ex line 201; 209; 255).

The methodological design is clear and appropriate to the study purpose and objectives. Ethical issues should reveal greater detail and be better described. The Scales are well presented and described.

The results are well represented in a simple way and illustrate the finding well. At this point it is considered that the analysis of each table must precede it, and not for the set of two as presented.

The discussion is well structured and clear. At this point, all the results of interest for the investigation were obtained and the comparison with other studies is good and current. They present the implications and limitations of this study and the importance of continuing to investigate this subject.

​The conclusions respond to the objective of the work, are synthetic and clearly illustrate to the reader the gains of this study. They highlight the gains for care practices and quality in health and academia, providing strategies for teachers and nursing curricula.

Aspects that need revision in the text:

Line 201 and line 209 (xxx); line 234-235 confirm the data; line 255 table title consistent with the data;

Best regards,

Author Response

Very interesting topic at the moment, well justified and well supported by scientific evidence in the introduction. The topic due to the positive impact it has on the academic life of nursing students and on the practice of quality nursing care.

Thank you for your comment.

Overall, the work is well written, with adequate scientific and research language and well-founded and despite the scarcity of research in the area, with good support.

Thank you for your comment.

References prior to 2013 (which appear in the introduction) do not benefit the work and can be removed with the exception of those associated with the scales used (2002 and 2006).

All references prior to 2013 were removed, but only major study, except the references associated with the scales, Watson (2008) as it defines a major concept in nursing “caring”, and (Mayer et al., 2008) as also defines the meaning of emotional intelligence.

There are some small details in the text that have to be revised, due to lack of text or information (marked lines for the authors, ex line 201; 209; 255).

Thank you for your concise comment. The texts were edited and highlighted.

The methodological design is clear and appropriate to the study purpose and objectives.

Thank you for your comment.

Ethical issues should reveal greater detail and be better described. The Scales are well presented and described

We added more information into the ethical considerations section: “Institutional Review Board (IRB) approval (Registration Number: 20-0494) was obtained from Princess Nourah bint Abdulrahman University IRB prior to the initiation of the investigation. Student nurses were provided with detailed information about the study and participation prior to obtaining consent. All students were informed that participation in the study is entirely voluntary, and the survey is anonymous. They were also assured that the study also has no potential for harm on their participation. They were also assured that the data would be secured and only be used for research purposes.  They were reassured that they can abstain from answering any questions that make them feel uncomfortable and withdraw from the study at any time without any penalty.”

The results are well represented in a simple way and illustrate the finding well. At this point it is considered that the analysis of each table must precede it, and not for the set of two as presented.

Thank you for your comment. We structured the presentation of results such that the analysis of each table precedes its display, contrary to the original presentation where the set of two tables was included in the manuscript without prior analysis.

The discussion is well structured and clear. At this point, all the results of interest for the investigation were obtained and the comparison with other studies is good and current. They present the implications and limitations of this study and the importance of continuing to investigate this subject.

Thank you for your comment

​The conclusions respond to the objective of the work, are synthetic and clearly illustrate to the reader the gains of this study. They highlight the gains for care practices and quality in health and academia, providing strategies for teachers and nursing curricula.

Thank you for your comment

Line 201 and line 209 (xxx); line 234-235 confirm the data; line 255 table title consistent with the data;

All texts were modified. 

Round 2

Reviewer 1 Report

Comments and Suggestions for Authors

The authors declare that they have changed the title, but the title has not been changed, as required.

The sample size and the ways in which it is representative of the entire reference population are not clear.

The authors did not properly implement the discussion as required.

Comments on the Quality of English Language

Moderate editing of English language required

Author Response

First reviewer’s comments

Response

The authors declare that they have changed the title, but the title has not been changed, as required.

Thanks for your comment. We apologize that we missed including that information. The territorial area was added “in Riyadh City, Saudi Arabia”

The sample size and the ways in which it is representative of the entire reference population are not clear.

Thanks for your comment. We clearly described the required sample size under the “Design, Setting, and Sample” section. We stated: “To ensure an adequate sample size for statistical power, a power analysis was calculated using G*Power. For the 6 predictor variables for the multiple linear regressions on the nurses’ caring behaviors, a sample of 146 student nurses was needed with a .15 effect size, .05 margin of error, and 95% statistical power.”

We also made it clear that our intent is to emphasize that the findings should not be generalized to the broader population in Saudi Arabia and beyond. The sample is not representative of the entire reference population, and caution should be exercised in interpreting the results without making broad generalizations. We stated: “As all participants were Saudi nursing students and recruited from one institution in one city in Saudi Arabia, the findings from this study have limited generalizability to other institutions and cities in Saudi Arabia. Furthermore, the sample predominantly comprises female nurses and as a result, caution should be exercised when attempting to generalize findings beyond the specific demographic characteristics of the study sample.”

The authors did not properly implement the discussion as required.

Thank you for your comment. The discussion section was revised.

Reviewer 2 Report

Comments and Suggestions for Authors

The paper was improved satisfactorily. There are typos and inconsistencies in using zeros before full stops in numbers. Please check the journal's guidelines.

Author Response

The paper was improved satisfactorily. There are typos and inconsistencies in using zeros before full stops in numbers. Please check the journal's guidelines.

Thank you for your comment. All zeros before the full stops in numbers “0.” were removed.
